# Exact mass analysis of sulfur clusters upon encapsulation by a polyaromatic capsular matrix

Sho Matsuno[1], Masahiro Yamashina[1,2], Yoshihisa Sei[1], Munetaka Akita[1], Akiyoshi Kuzume[1], Kimihisa Yamamoto[1] & Michito Yoshizawa[1]

Structural determination of inorganic clusters relies heavily on mass spectrometry because of, in most cases, their poor responsivities toward nuclear magnetic resonance, ultraviolet/visible, and infrared analyses. Nevertheless, mass spectrometry analysis of oligosulfurs ($S_n$), which are unique clusters with copious allotropic forms, usually displays their fragment peaks. Here we report that a polyaromatic capsule acts as a supramolecular matrix for the mass determination of the neutral sulfur clusters. Upon encapsulation, molecular ion peaks derived from the host–guest complexes including cyclic $S_6$ and $S_8$ clusters are exclusively detected by common electrospray ionization time-of-flight mass spectrometry analysis. Furthermore, mass spectrometry analysis of a cyclic $S_{12}$ cluster, which is in situ prepared from two $S_6$ clusters within the matrix upon light irradiation, is achieved by the same way. The present matrix can remarkably stabilize the otherwise labile $S_6$ and $S_{12}$ clusters in the polyaromatic shell not only under mass spectrometry conditions but also in an ambient solution state.

---

[1] Laboratory for Chemistry and Life Science, Institute of Innovative Research, Tokyo Institute of Technology, 4259 Nagatsuta, Midori-ku, Yokohama 226-8503, Japan. [2] Present address: Department of Chemistry, University of Cambridge, Lensfield Road, Cambridge CB2 1EW, UK. Correspondence and requests for materials should be addressed to M.Yoshizawa. (email: yoshizawa.m.ac@m.titech.ac.jp)

Mass spectrometry (MS) is of particular importance for the structural characterization of inorganic clusters, because more precise information is not easily obtained by other analytical techniques such as nuclear magnetic resonance (NMR), ultraviolet-visible (UV/Vis), and infrared (IR) spectroscopies[1, 2]. However, most of the neutral clusters fully or partially decompose within MS[3], which prevents us from determining the molecular weight of the target structures. Sulfur clusters continue to attract attention in the fields of physical and synthetic chemistry due to their unique structures and reactivities[4–7]. More than 30 allotropes of sulfur (e.g., $S_{6-15}$) are known to date, yet mass determination of the uncharged structures has been virtually impossible by previous MS methods. Owing to the instability of oligosulfur ions generated under usual electron ionization conditions, immediate and substantial fragmentation reactions are typically observed (Fig. 1a, left). Therefore, no reliable method to obtain the structural information of such metastable clusters in solution has been reported. We anticipated that if the neutral sulfur clusters (Fig. 1b) were fully wrapped with an ionic capsular matrix, the exact molecular weights could be readily available on the MS analysis (Fig. 1a, right), because of the protection of the encapsulated clusters and the promotion of the efficient ionization. In addition, development of a new class of supramolecular matrixes for common mass spectrometry would assist in the creation of not only unknown sulfur clusters but also novel metalloclusters. There are many

reports on the successful stabilization and observation of highly reactive organic and organometallic compounds within supramolecular cages and capsules[8–18]. Nevertheless, those of labile inorganic clusters are limited to white phosphorus (Fig. 1c)[19] and yellow arsenic[20, 21].

Here we report the MS analysis of cyclic sulfur clusters $S_n$ ($n = 6$, 8, and 12; Fig. 1b) upon encapsulation. As an analytical matrix, we employ supramolecular capsule 1 (Fig. 1d)[22] bearing a positively charged (4+) polyaromatic shell suitable for electrospray ionization time-of-flight (ESI-TOF) MS analysis[2]. The composition of the supramolecular matrix is simple, i.e., $M_2L_4$ with two Pd(II) ions and four bent bispyridine ligands with two anthracene panels[23]. The spherical cavity surrounded by the multiple polyaromatic panels can efficiently accommodate various organic molecules with dimensions of up to 1 nm (e.g., BODIPY, corannulene, and fullerene $C_{60}$) in aqueous solutions through the hydrophobic effect and π-stacking interactions[18, 24–26]. However, solution-state host–guest interactions between the inorganic sulfur clusters and our capsules as well as other supramolecular cages remained uncertain so far[27, 28]. In the present work, we demonstrate the exact mass determination of cyclic $S_6$ and $S_8$ clusters by common ESI-TOF MS analysis upon encapsulation within capsular matrix 1. In addition, we accomplish selective preparation of a cyclic $S_{12}$ cluster from two $S_6$ clusters within the matrix upon light irradiation, as revealed by subsequent MS analysis of the host–guest complex.

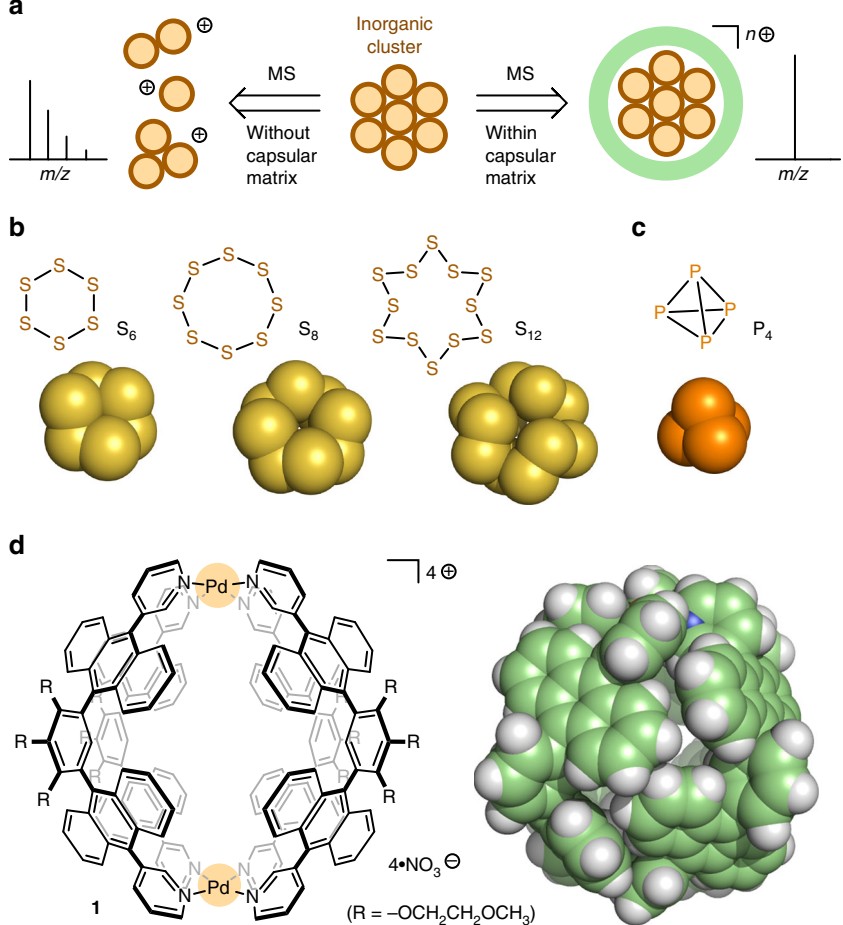

**Fig. 1** MS analysis of an inorganic cluster and structure of sulfur clusters and polyaromatic capsule **1**. **a** Schematic representation of the MS analysis of a neutral inorganic cluster within/without a capsular matrix and their typical ion peaks. **b** Sulfur clusters $S_6$, $S_8$, and $S_{12}$ and their crystal or optimized structures. **c** White phosphorus and its crystal structure[37]. **d** Polyaromatic capsule **1** and the space-filling representation of the crystal structure (substituents and counterions are omitted for clarity)

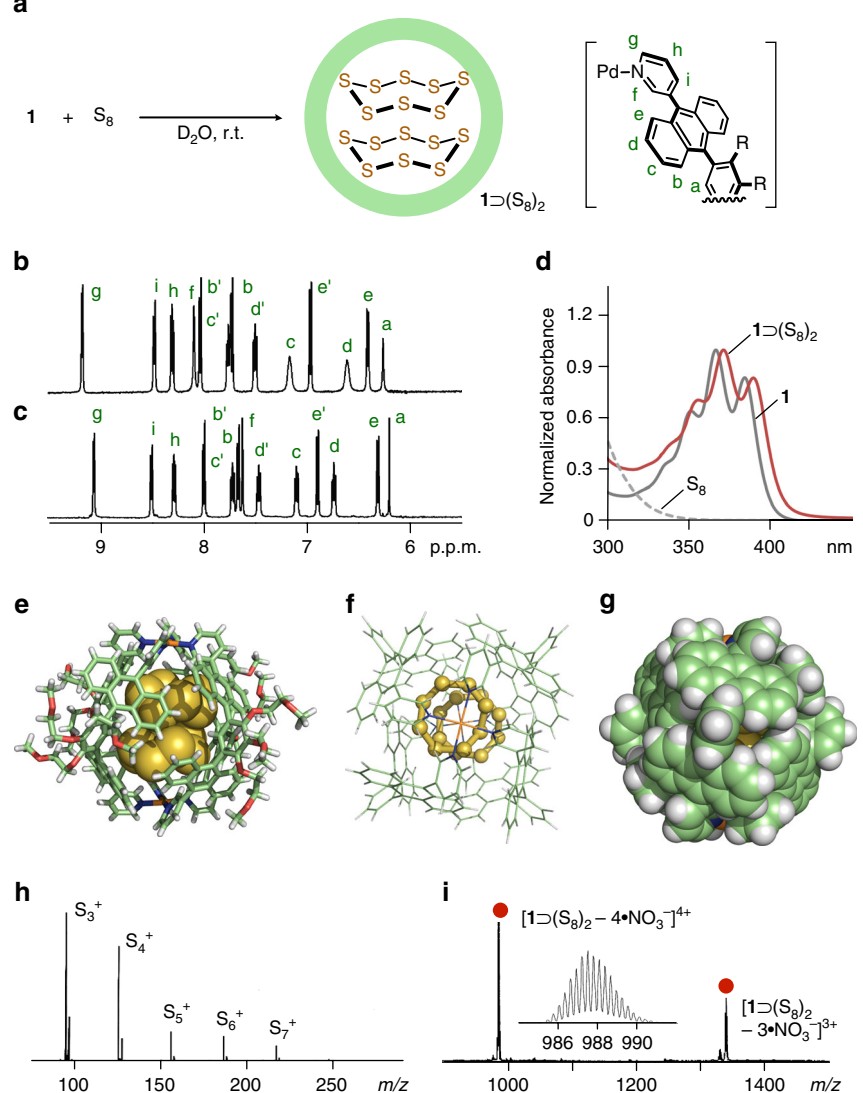

**Fig. 2** Formation and characterization of **1**⊃(S$_8$)$_2$. **a** Schematic representation of the encapsulation of two S$_8$ clusters within matrix **1**. $^1$H NMR spectra (500 MHz, D$_2$O, room temperature) of **b 1** and **c 1**⊃(S$_8$)$_2$. **d** UV-visible spectra (room temperature) of **1** and **1**⊃(S$_8$)$_2$ in H$_2$O, and S$_8$ in CH$_3$OH. X-ray crystal structure of **1**⊃(S$_8$)$_2$ (host and guest parts): **e** cylinder and space-filling representation, **f** ball-and-stick, and **g** space-filling representations (the peripheral substituents are replaced by hydrogen atoms). **h** MALDI-TOF MS spectrum of S$_8$. **i** ESI-TOF MS spectrum (H$_2$O) of **1**⊃(S$_8$)$_2$ and the expansion of the [**1**⊃(S$_8$)$_2$−4•NO$_3^-$]$^{4+}$ peak

## Results

**MS analysis of a cyclic S$_8$ cluster**. We firstly carried out the quantitative encapsulation and mass determination of *cyclo*-octasulfur (S$_8$) using capsular matrix **1** in water. Stirring excess hydrophobic S$_8$ (3.9 μmol) in a D$_2$O solution (0.5 ml) of **1** (0.40 μmol) at room temperature for 30 min led to the exclusive formation of a **1**⊃(S$_8$)$_2$ complex (Fig. 2a). After removal of suspended free S$_8$ by filtration, the 1:2 host–guest structure was confirmed by NMR, UV-visible, and X-ray crystallographic analyses. In the $^1$H NMR spectrum, the appearance of new aromatic signals and the disappearance of the original matrix signals are indicative of the quantitative encapsulation of the sulfur cluster (Figs. 2b, c and see Supplementary Fig. 1). Due to the inclusion, the internal $H_f$ signal of **1** was largely shifted upfield ($\Delta\delta = -0.40$ p.p.m.), whereas the external $H_h$ signal remained almost unchanged. UV-visible spectrum of the product showed slight red-shifts ($\Delta\lambda_{max} = +6$ nm) of the absorption bands derived from the anthracene moieties of **1** (Fig. 2d), indicating S-$\pi$ interactions[29]. Direct evidence of the **1**⊃(S$_8$)$_2$ structure was

obtained from the X-ray crystallographic analysis. Pale-yellow crystals suitable for X-ray analysis grew by the slow evaporation of a 25:1 H$_2$O/CH$_3$CN solution of the product at room temperature for 1 month. The molecular structure revealed that two molecules of S$_8$ fully occupy the spherical cavity of **1** in a stacked fashion (Figs. 2e, f and see Supplementary Fig. 3). Each of the crown-shaped rings is in close proximity with the closest intermolecular S•••S distance of 3.1 Å. With regard to the host–guest interactions, the closest distances between the sulfur atoms and the eight anthracene panels are less than 3.6 Å (see Supplementary Fig. 4), whereas those between the sulfur atoms and the Pd (II) centers are more than 4.8 Å. The S$_8$ dimer can be isolated from the bulk phase by the multiple anthracene panels of **1** (Fig. 2g).

Although the S$_8$ cluster is the most stable sulfur allotrope[6], the matrix-assisted laser desorption ionization (MALDI)-TOF MS spectrum displays several ion peaks derived from the fragments such as S$_3^+$, S$_4^+$, and S$_5^+$ species ($m/z$ = 96, 128, and 160, respectively; Fig. 2h) even under various conditions[30]. ESI-TOF

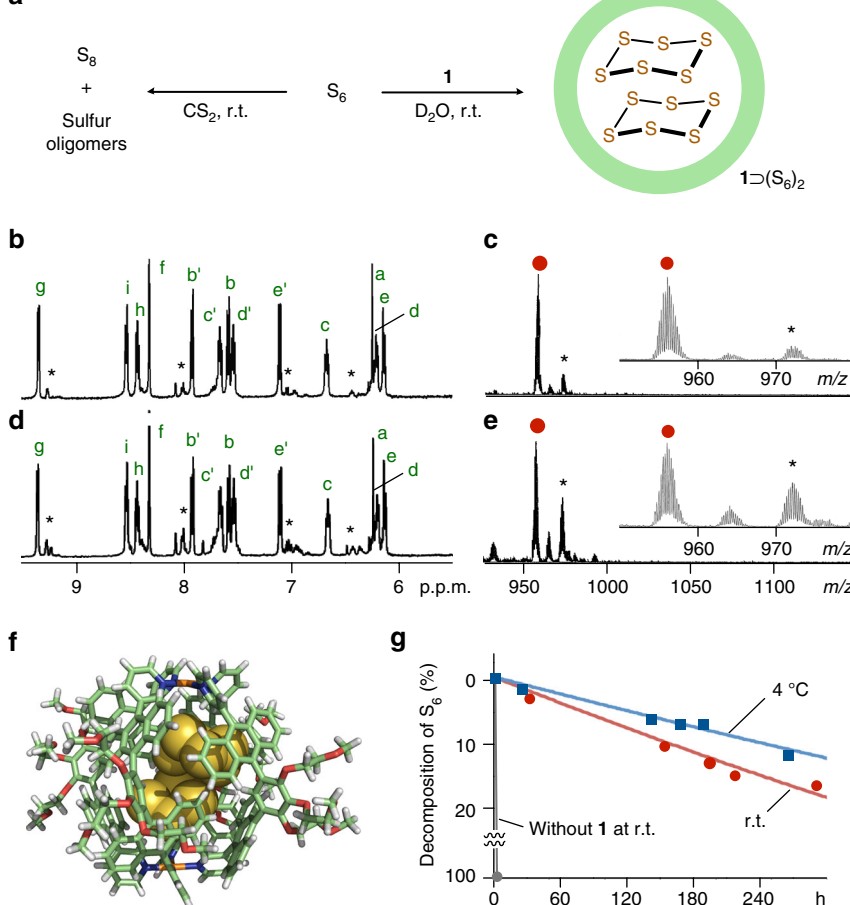

**Fig. 3** Formation and characterization of $1 \supset (S_6)_2$ and its stability. **a** Schematic representation of the encapsulation of two $S_6$ clusters within matrix **1** in $D_2O$ (*right*) and the decomposition of $S_6$ in $CS_2$ (*left*). [1]H NMR (500 MHz, $D_2O$, room temperature; *left*) and ESI-TOF MS spectra ($H_2O$; *right*) of $1 \supset (S_6)_2$ **b, c** before and **d, e** after 8 days at room temperature (*asterisks* indicate signals derived from $1 \supset (S_6 \bullet S_8)$). **f** X-ray crystal structure of $1 \supset (S_6)_2$: cylindrical and space-filling representation (solvents and counterions are omitted for clarity). **g** Decomposition of $S_6$ (%) within matrix **1** in water and without matrix **1** in $CS_2$

MS analysis of $S_8$ also shows no target peak (see Supplementary Fig. 5). In contrast, ESI-TOF MS spectrum of the product showed prominent peaks at $m/z = 988.0$ and 1338.1, corresponding to 1:2 host–guest $[1 \supset (S_8)_2 - n \bullet NO_3^-]^{n+}$ species ($n = 4$ and 3, respectively; Fig. 2i and see Supplementary Fig. 6). No MS peaks assignable to the empty matrix and host–guest complexes including decomposed $S_8$ clusters were detected so that the significant stabilization of $S_8$ under MS conditions was demonstrated upon encapsulation within **1**. The non-covalent host–guest structure remains intact in water even under highly diluted conditions (<5.0 μM; see Supplementary Fig. 7), suggesting the potential application for microgram-scale MS analysis. Interestingly, the binding is stronger than that of hydrophobic cyclooctane. [1]H NMR competitive binding experiments revealed that the matrix encapsulates $S_8$ clusters with >90% selectivity from a 1:1 mixture of the cyclic sulfur and alkane (20 equiv. each) under ambient aqueous conditions (see Supplementary Fig. 8).

**Stabilization and MS analysis of a cyclic $S_6$ cluster**. To investigate how capsular matrix **1** impacts the MS analysis of a metastable sulfur allotrope, we next synthesized pure *cyclo*-hexasulfur ($S_6$) at low temperatures and prepared the host–guest complex[31]. In a manner similar to the encapsulation of $S_8$, simple mixing hydrophobic $S_6$ (13 equiv.) with **1** in $D_2O$ gave rise to a $1 \supset (S_6)_2$ complex predominantly (Fig. 3a). A [1]H NMR spectrum

of the resultant solution displayed new prominent signals in the aromatic region derived from the desired product (Fig. 3b and see Supplementary Fig. 10). One set of minor signals was also found at, e.g., 9.27, 8.07, and 8.01 p.p.m., which suggests the formation of a $1 \supset (S_6 \bullet S_8)$ complex (<10%) due to the decomposition of $S_6$ during the mixing. UV-visible absorption bands of **1** around 390 nm were again slightly red-shifted ($\Delta \lambda_{max} = +3$ nm) after treatment with the $S_6$ solid (see Supplementary Fig. 11). The encapsulation of two cyclic $S_6$ clusters within the matrix was preliminarily confirmed by the X-ray crystallographic analysis (Fig. 3f and see Supplementary Figs 12 and 13) and yet definitely proved by the ESI-TOF MS analysis. Prominent molecular ion peaks were observed in the spectrum at $m/z = 956.0$, 1295.4, and 1974.6, derived from $[1 \supset (S_6)_2 - n \bullet NO_3^-]^{n+}$ species ($n = 4$, 3, and 2, respectively; Fig. 3c). The isotope patterns of the peaks fully coincide with the calculated ones (see Supplementary Fig. 14). As expected, minor MS peaks found at $m/z = 972.0$ and 1317.0 were assignable to a $1 \supset (S_6 \bullet S_8)$ species. In contrast to the host–guest complex, without the matrix, the MALDI-TOF MS analysis of free $S_6$ showed only the fragment peaks (see Supplementary Fig. 15).

It is worth noting that the otherwise labile $S_6$ cluster, which possesses a highly strained ring structure[4, 6, 32], is remarkably stabilized within matrix **1** not only under MS conditions but also in an ambient solution state[33]. Free $S_6$ dissolved in $CS_2$ was completely decomposed into $S_8$ and other oligosulfurs within ~1

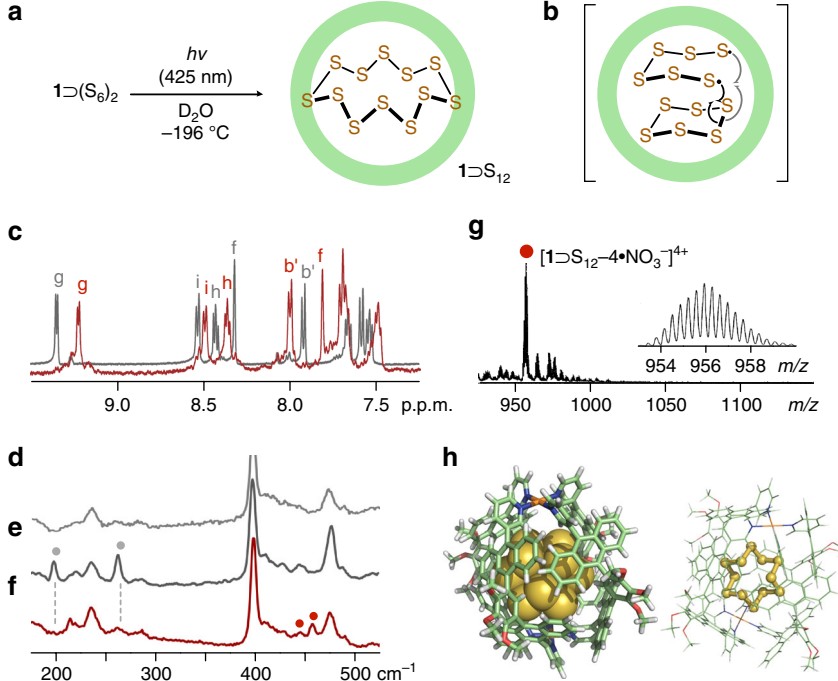

**Fig. 4** Preparation and characterization of **1**⊃S$_{12}$. **a** Schematic representation of the transformation from two S$_6$ clusters into a S$_{12}$ cluster within matrix **1** upon light irradiation and **b** the proposed intermediate. **c** $^1$H NMR spectra (500 MHz, D$_2$O, room temperature) of **1**⊃S$_{12}$ (*red line*) and **1**⊃(S$_6$)$_2$ (*gray line*). Raman spectra (He−Ne laser, $\lambda_{ex}$ = 632.8 nm, room temperature) of **d 1**, **e 1**⊃(S$_6$)$_2$, and **f 1**⊃S$_{12}$. **g** ESI-TOF MS spectrum (H$_2$O) of **1**⊃S$_{12}$. **h** Optimized structure of **1′**⊃S$_{12}$ (R = −OCH$_3$): cylindrical and space-filling representation (*left*) and ball-and-stick representation (*right*)

h at room temperature (Fig. 3g and see Supplementary Fig. 17). In sharp contrast, time-dependent $^1$H NMR analysis revealed that 75% of the proton signals of **1**⊃(S$_6$)$_2$ remain unchanged after 8 days (Fig. 3d and see Supplementary Fig. 18) and the ESI-TOF MS analysis of the resultant solution indicated the generation of **1**⊃(S$_6$•S$_8$) and **1**⊃(S$_6$•S$_7$) species (Fig. 3e). The $\tau_{1/2}$ of the S$_6$ clusters within **1** was estimated to be 770 h by $^1$H NMR studies (Fig. 3g). The decomposition is further suppressed at lower temperatures (e.g., $\tau_{1/2}$ = 1733 h at 4 °C). The observed, unusual stabilization of encapsulated S$_6$ clusters in solution as well as under the MS conditions presumably arises from the isolation effect of the polyaromatic shell of matrix **1**.

**In situ preparation and MS analysis of a cyclic S$_{12}$ cluster**. It is known that sulfur clusters are photosensitive and, in most cases, rapidly decomposed into complex mixtures in solutions upon light irradiation[4–6]. Notably, selective photochemical transformation from two S$_6$ clusters into a cyclic S$_{12}$ cluster and subsequent MS analysis of the product were attained using matrix **1**. When a frozen aqueous glass of **1**⊃(S$_6$)$_2$ cooled at −196 °C was irradiated by LED lamps (3 W × 4, $\lambda_{irrd}$ = 425 ± 15 nm) for 30 min, distinct shifts of the matrix signals were observed in the NMR spectrum (Figs. 4a, c and see Supplementary Fig. 19). The signal pattern is quite different from that of empty **1** and host–guest complex **1**⊃(S$_8$)$_2$, elucidating the formation of a new photoproduct within **1**. In addition, a huge upfield shift of inner protons $H_f$ is indicative of the encapsulation of a bulky cluster. The product could not be extracted with CS$_2$ solutions, because of most probably its steric demand. This photoreaction occurred very slowly under UV light irradiation ($\lambda_{irrd}$ = 360 nm), probably due to the strong absorption bands (~380 nm) of the polyaromatic capsule (see Supplementary Fig. 20). A Raman spectrum of the product supported the formation of the cyclic S$_{12}$ cluster (Figs. 4d, f and see Supplementary Fig. 21). The observed new peaks at 445 and 457 cm$^{-1}$ are overlapped with the characteristic

peaks of free S$_{12}$ (i.e., 445 and 456 cm$^{-1}$, respectively)[34]. On the other hand, Raman peaks derived from clusters S$_6$ (e.g., 200 and 264 cm$^{-1}$; Fig. 4e) and S$_8$ (e.g., 149 and 218 cm$^{-1}$) within **1** were virtually undetected in the spectrum. These peaks are slightly shifted (up to −4 cm$^{-1}$) as compared with those of free S$_6$ and S$_8$ (see Supplementary Figs 16 and 9). Finally, ESI-TOF MS spectrum of the product exhibited intense peaks at $m/z$ = 956.0 and 1295.7, assigned to 1:1 host–guest [**1**⊃S$_{12}$−$n$•NO$_3^-$]$^{n+}$ species ($n$ = 4 and 3, respectively; Fig. 4g and see Supplementary Fig. 22). The MS peaks for **1** and **1**⊃(S$_8$)$_2$ were not observed in the spectrum. The optimized structure of **1′**⊃S$_{12}$ (R = −OCH$_3$) by force-field calculations displays that the spherical S$_{12}$ cluster with a diameter of 0.66 nm is fully insulated by the spherical polyaromatic framework of **1′** (Fig. 4h and see Supplementary Fig. 23). The otherwise reactive S$_{12}$ cluster in the capsular shell remains in more than 60% for 8 days at room temperature (see Supplementary Figs 24 and 25). Concerning the unusual clusterization in the isolated nanospace, the S$_6$ cluster provides a highly strained, small ring structure, whereas the product is the next stable sulfur allotrope after S$_8$. Thus, the formation of the S$_{12}$ cluster through the generation of a biradical species (Fig. 4b), by homolytic photocleavage of the S–S bond of the encapsulated S$_6$, might be a thermodynamically favorable process[32, 35].

## Discussion
We have succeeded in the facile MS characterization of inorganic sulfur clusters S$_6$, S$_8$, and S$_{12}$ by using a polyaromatic capsule as a new supramolecular matrix, whereas the clusters themselves fragment under usual MS conditions. Simple mixing cyclic S$_6$ or S$_8$ clusters with the capsular matrix provides stable 1:2 host–guest complexes for common ESI-TOF MS analysis. The cationic and closed polyaromatic shell of the matrix facilitates the efficient ionization of the uncharged clusters without troublesome fragmentation. In addition, the capsular matrix can be used for not only in situ synthesis but also MS analysis of a cyclic S$_{12}$ cluster.

The otherwise labile $S_6$ and $S_{12}$ clusters are also significantly stabilized within the matrix in an ambient solution state. The present functions as an analytical tool as well as a reaction vessel for sulfur clusters prompt us to discover unknown inorganic clusters by using the polyaromatic capsular matrix and its potential derivatives[36].

## Methods

**General**. NMR: Bruker ASCEND-500 (500 MHz), ESI-TOF MS: Bruker micrO-TOF II, FT-IR: JASCO FT/IR-4200, X-ray: Bruker AXS D8 VENTURE/PHOTON 100 diffractometer Raman: JASCO NRS-4100 or HORIBA, Ltd LabRAM HR Evolution. Optimized structure: Materials Studio (ver. 5.5.3). Solvents and reagents: TCI Co., Ltd, Wako Pure Chemical Industries Ltd, Kanto Chemical Co., Inc., Sigma-Aldrich Co., and Cambridge Isotope Laboratories, Inc. Polyaromatic capsule **1** and sulfur cluster $S_6$ (see Supplementary Methods) were synthesized according to previously reported procedures[18, 31].

**Raman analysis**. The excitation wavelength from an He-Ne laser was 632.8 nm with a power on the sample typically 22 μW. A long-working distance objective (× 50 magnification, 10 mm focal length) was used to focus the laser onto the sample. The Raman signal was collected in a back-scattering geometry.

**Synthesis of $1 \supset (S_8)_2$**. Capsular matrix **1** (1.5 mg, 0.40 μmol), $S_8$ (1.0 mg, 3.9 μmol), and $D_2O$ (0.5 ml) were added to a microtube (2 ml) and the resultant mixture was stirred at room temperature for 30 min. The quantitative formation of a $1 \supset (S_8)_2$ complex was confirmed by NMR (see Supplementary Figs 1 and 2), X-ray crystallographic (see Supplementary Figs 3 and 4 and Supplementary Table 1), ESI-TOF MS (see Supplementary Fig. 6), and Raman (see Supplementary Fig. 9) analyses. The host–guest structure of $1 \supset (S_8)_2$ is stable enough in water at room temperature even under high dilution conditions (5.0 μM), as confirmed by $^1H$ NMR analysis (see Supplementary Fig. 7).

$^1H$ NMR (500 MHz, $D_2O$, room temperature): $\delta$ 2.40 (s, 24H), 3.00 (m, 16H), 3.44 (s, 12H), 3.90 (m, 16H), 4.02 (m, 8H), 4.42 (m, 4H), 4.60 (m, 4H), 6.27 (s, 4H), 6.38 (d, $J = 8.5$ Hz, 8H), 6.80 (dd, $J = 8.5$, 7.5 Hz, 8H), 6.96 (d, $J = 8.5$ Hz, 8H), 7.16 (dd, $J = 8.5$, 7.5 Hz, 8H), 7.52 (dd, $J = 8.5$, 7.5 Hz, 8H), 7.68 (d, $J = 8.5$ Hz, 8H), 7.72 (d, $J = 8.5$ Hz, 8H), 7.78 (dd, $J = 8.5$, 7.5 Hz, 8H), 8.05 (d, $J = 8.5$ Hz, 8H), 8.33 (dd, $J = 8.0$, 5.5 Hz, 8H), 8.56 (d, $J = 8.0$ Hz, 8H), 9.10 (d, $J = 5.5$ Hz, 8H). $^{13}C$ NMR (125 MHz, $D_2O$, room temperature): $\delta$ 57.0 ($CH_3$), 58.1 ($CH_3$), 70.6 ($CH_2$), 71.7 ($CH_2$), 72.6 ($CH_2$), 72.7 ($CH_2$), 124.2 (CH), 124.8 (CH), 126.2 (CH), 126.5 (CH), 127.0 (CH), 127.2 (CH), 127.3 (CH), 127.7 ($C_q$), 128.1 ($C_q$), 128.4 (CH), 129.3 ($C_q$), 129.4 ($C_q$), 129.5 ($C_q$), 129.8 (CH), 134.9 ($C_q$), 138.4 ($C_q$), 145.3 (CH), 145.6 ($C_q$), 151.5 ($C_q$), 151.7 (CH), 152.1 (CH). Raman ($\lambda_{ex} = 632.8$ nm, 22 μW, $cm^{-1}$): 149, 218, 238, 400, 478. ESI-TOF MS ($H_2O$): $m/z$ 2038.6 $[1 \supset (S_8)_2 - 2 \bullet NO_3^-]^{2+}$, 1338.1 $[1 \supset (S_8)_2 - 3 \bullet NO_3^-]^{3+}$, 988.0 $[1 \supset (S_8)_2 - 4 \bullet NO_3^-]^{4+}$.

**Synthesis of $1 \supset (S_6)_2$**. Capsular matrix **1** (1.5 mg, 0.40 μmol), $S_6$ (1.0 mg, 5.1 μmol), and $D_2O$ (0.5 ml) were added to a microtube (2 ml). The mixture was stirred at room temperature for 15 min. The selective formation of a $1 \supset (S_6)_2$ complex (>90% yield) was confirmed by NMR (see Supplementary Fig. 10), UV-visible (see Supplementary Fig. 11), X-ray crystallographic (see Supplementary Figs 12 and 13 and Supplementary Table 2), ESI-TOF MS (see Supplementary Fig. 14), and Raman (see Supplementary Fig. 16) analyses.

$^1H$ NMR (500 MHz, $D_2O$, room temperature): $\delta$ 2.39 (s, 24H), 3.00 (m, 16H), 3.43 (s, 12H), 3.82 (m, 8H), 3.89 (m, 8H), 3.96 (m, 8H), 4.39 (m, 4H), 4.59 (m, 4H), 6.14 (d, $J = 8.5$ Hz, 8H), 6.21 (dd, $J = 8.5$, 7.5 Hz, 8H), 6.25 (s, 4H), 6.67 (dd, $J = 8.5$, 7.5 Hz, 8H), 7.11 (d, $J = 8.5$ Hz, 8H), 7.54 (dd, $J = 8.5$, 7.5 Hz, 8H), 7.59 (d, $J = 8.5$ Hz, 8H), 7.66 (dd, $J = 8.5$, 7.5 Hz, 8H), 7.92 (d, $J = 8.5$ Hz, 8H), 8.32 (s, 8H), 8.43 (dd, $J = 8.0$, 6.0 Hz, 8H), 8.53 (d, $J = 8.0$ Hz, 8H), 9.35 (d, $J = 6.0$ Hz, 8H). Raman ($\lambda_{ex} = 632.8$ nm, 22 μW, $cm^{-1}$): 200, 238, 264, 398, 478. ESI-TOF MS ($H_2O$): $m/z$ 1974.6 $[1 \supset (S_6)_2 - 2 \bullet NO_3^-]^{2+}$, 1295.4 $[1 \supset (S_6)_2 - 3 \bullet NO_3^-]^{3+}$, 956.0 $[1 \supset (S_6)_2 - 4 \bullet NO_3^-]^{4+}$.

**Formation of $1 \supset S_{12}$**. An aqueous solution of $1 \supset (S_6)_2$ (0.8 mM, 0.5 ml) was irradiated with Relyon LED lamps (3 W × 4; $\lambda_{irrd} = 425 \pm 15$ nm) for 30 min at $-196$ °C (under liquid $N_2$). The selective formation of a $1 \supset S_{12}$ complex was confirmed by $^1H$ NMR (see Supplementary Fig. 19), UV-visible (see Supplementary Fig. 20), Raman (see Supplementary Fig. 21), and ESI-TOF MS (see Supplementary Fig. 22) analyses. On the other hand, light irradiation (3 W × 2) of $1 \supset (S_6)_2$ in $D_2O$ for 2 h at room temperature gave rise to complex mixtures.

$^1H$ NMR (500 MHz, $D_2O$, room temperature): $\delta$ 2.39 (s, 24H), 3.02 (m, 16H), 3.45 (s, 12H), 3.92 (m, 16H), 3.99 (m, 8H), 4.44 (m, 4H), 4.61 (m, 4H), 6.15 (d, $J = 8.5$ Hz, 8H), 6.36 (dd, $J = 8.5$, 7.0 Hz, 8H), 6.49 (s, 4H), 6.92 (dd, $J = 8.5$, 7.0 Hz, 8H), 7.01 (d, $J = 8.5$ Hz, 8H), 7.49 (dd, $J = 8.5$, 7.0 Hz, 8H), 7.72-7.69 (m, 16H), 7.82 (s, 8H), 8.00 (d, $J = 8.5$ Hz, 8H), 8.36 (dd, $J = 7.5$, 5.5 Hz, 8H), 8.49 (d, $J = 7.5$ Hz, 8H), 9.22 (d, $J = 5.5$ Hz, 8H). Raman ($\lambda_{ex} = 632.8$ nm, 22 μW, $cm^{-1}$): 180, 216, 238, 263, 288, 398, 410, 445, 457, 475, 490. ESI-TOF MS ($H_2O$): $m/z$ 1974.0 $[1 \supset S_{12} - 2 \bullet NO_3^-]^{2+}$, 1295.7 $[1 \supset S_{12} - 3 \bullet NO_3^-]^{3+}$, 956.0 $[1 \supset S_{12} - 4 \bullet NO_3^-]^{4+}$.

**Data availability**. The authors declare that the data supporting the findings of this study are available within the Supplementary Information files and from the corresponding author upon reasonable request. CCDC-1509515 and CCDC-1525707 contain the supplementary crystallographic data for the structures reported in this article. These data can be obtained free of charge from The Cambridge Crystallographic Data Centre (CCDC) via www.ccdc.cam.ac.uk/data_request/cif.

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

## Acknowledgements

This work was supported by JSPS KAKENHI (Grant No. JP25104011/JP26288033/JP17H05359/JP15H05757) and "Support for Tokyotech Advanced Researchers (STAR)". M.Ya. thanks the JSPS for an Overseas Research Fellowship. A.K. and K.Y. acknowledge the financial support from JST ERATO (Grant No. JPMJER1503).

## Author contributions

S.M., M.Ya., and M.Yo. designed the work, carried out research, analyzed data, and wrote the paper. Y.S., M.A., A.K., and K.Y. were involved in the work discussion. Y.S. and A.K. contributed to X-ray crystallographic and Raman analyses, respectively. M.Yo. is the principal investigator. All authors discussed the results and commented on the manuscript.

## Additional information

**Competing interests:** The authors declare no competing financial interests.

