## [Peer Review File-New · Nature Communications]

Reviewer #1 (Remarks to the Author):

The manuscript entitled "A Polyaromatic Capsular Matrix: Exact Mass Analysis of Sulfur Clusters upon Encapsulation" by Yoshizawa coworkers reported the stability of some Sn clusters (S6, S8) under ESI-MS condition when they are encapsulated by a Pd-cage. This manuscript cannot be recommended to Nature Commun for publication due to the following literature precedence and reviewer's analysis.

- Synthesis of the cage was previously demonstrated by the authors (N. Kishi, Z. Li, K. Yoza, M. Akita and M. Yoshizawa, *J. Am. Chem. Soc.*, 2011, 133, 11438-11441). Following, encapsulation of certain disulfide molecules by the same cage is also reported by them (N. Kishi, Z. Li, Y. Sei, M. Akita, K. Yoza, J. S. Siegel and M. Yoshizawa, *Chem. Eur. J.*, 2013, 19, 6313-6320).
- Stabilization of reactive species by the same cage. Authors have already shown radical initiator AIBN ion could be stabilized towards light and heat by the same Pd-Cage (M. Yamashina, Y. Sei, M. Akita and M. Yoshizawa, *Nature Commun.*, 2014, 5, 4662).
- Literature on encapsulation of S8. Raptis and co-workers have reported the crystal structure and stability of S8 with trigonal prismatic cage molecules (P.-C. Duan, Z.-Y. Wang, J.-H. Chen, G. Yang and R. G. Raptis, *Dalton Trans.*, 2013, 42, 14951-14954).
- Literature on stabilization of S6. Sugimoto et al., reported a discovery of air-stable S6 molecules in the solid state via crystallization of co-crystals with 3,5-diphenyl-1,2,4-dithiazol-1-ium (dpdti) iodide (ref: K. Sugimoto, H. Uemachi, M. Maekawa and A. Fujiwara, *Cryst. Growth Des.*, 2013, 13, 433-436). The ring-structure cyclo-S6 molecule has been reported to be stored in cocrystals for at least half a year. Contrastingly authors have reported here identification of <10% of the cage \supset (S6•S8) complex at NMR time scale.

This findings can thus be rationalized as a routine extension to the concept of cage to stabilization of reactive species (references 8-19). As a reviewer, I am not convinced that the results signify a sufficiently notable advance to justify publication in Nature Communications.

Reviewer #2 (Remarks to the Author):

This well-written and well-illustrated manuscript describes the preparation and characterization of three supramolecular compounds (in tiny quantities) which should be highly interesting to both main-group and coordination chemists. In principle, water-insoluble elemental sulfur has been encapsulated in aqueous solution by a polyaromatic palladium complex 1 characterized by a large inner cavity which is big enough to host two S8 or two S6 molecules resulting in the crystalline species 1x(S8)2 and 1x(S6)2. These two species have been well characterized spectroscopically as well as by X-ray structural analysis. It is obvious that these findings will be the starting point for many subsequent studies on the encapsulation and stabilization of other hydrophobic inorganic molecules. A third compound claimed to be 1x(S12) has been obtained by low-temperature irradiation of 1x(S6)2. However, this compound has been less well characterized (no X-ray structure analysis) and should be either omitted from the manuscript or better characterized. Especially, since the Raman spectrum of 1x(S12) does not show the most intense line of S12 at 128 cm⁻¹. A better spectrum would be of great help to convince this reviewer.

Thus, I strongly recommend acceptance of the manuscript for publication after additional revisions as follows: correction of ref. 31 (two spelling errors). Also, 12 times the word "exact" is used in the manuscript which seems somewhat exaggerated since measurements should always be exact. Furthermore, MS is not a method for structural characterization of sulfur clusters (page 1) but just for mass determination; the most important spectroscopic technique for sulfur clusters is Raman spectroscopy which can also be applied to solutions (page 2). The closest distances between the sulfur atoms and the eight anthracene panels of 3.6 Å (page 4) hardly suggest the presence of any S- π interactions and the hydrophobic effect is most probably responsible alone for the formation of these encapsulated sulfur complexes. On p. 6 the authors should mention whether or not the slow decomposition of $1 \times (S_6)_2$ in aqueous solution results in the increase of signals due to $1 \cdot (S_6 \cdot S_8)$ or what other decomposition products have been observed; see Fig. 3e.

Reviewer #3 (Remarks to the Author):

The ms by Yoshizawa et al reports a very appealing mass spectrometry and single crystal X-ray diffraction (SCXRD) study on sulfur S_6 , S_8 and S_{12} clusters captured inside a nanocontainer. As the nanocontainer they have designed an ingeniously simple polyaromatic nanocapsule which is then used as a supramolecular matrix. The sulfur clusters fragment under usual mass spectrometry conditions, yet isolation and protecting the S-cluster inside the nanocontainer provides stable 1:2 host-guest complexes for a routine ESI-TOF MS analysis. The cationic and fully closed polyaromatic nanocontainer not only protects the clusters, but also facilitates the efficient ionization of the clusters without fragmentation. In addition, the capsular matrix shows a promise that they could be used as nanoreactors. The authors have also provided SCXRD proof about the encapsulated S-clusters, this is not an easy task yet they have done it remarkably well.

I am very happy to support the publication of this excellent piece of work in Nature Communications, provided that some minor matters in the checkcif files are commented, see below.

1. The checkcif report contains quite a bit of A and B level alerts. This is natural with such complex X-ray structures, yet the authors should comment these alerts using the IUCr Output Validation Response Form available from <http://checkcif.iucr.org/>

Reviewer #4 (Remarks to the Author):

This is a very interesting paper that reports on an interesting method that might be uniquely suited to study neutral main group clusters in the gas phase that would otherwise be charged and undergo thermodynamically induced mainly dismutation reactions.

Thus, I can support publication, yet only after the below addressed major concerns are addressed.

Remarks: Overall the authors use the study of neutral and charged gaseous clusters with absolutely no differentiation. This is not valid. The uniqueness of their method is that it allows to study neutral clusters

encapsulated into a matrix of an MOF-like environment. In addition, they give absolutely no thermodynamic reasoning for the reactivity observed (i.e. the dimerization of 2 S6 giving S12). However, this is a downhill reaction and only logic. Thus, the driving force needs to be accounted for.

I have addressed all those points following their appearance in the text in the following. Once those points are addressed, and suitably incorporated in a revised manuscript with re-review, the manuscript may become suitable for NCOMMS.

Nomenclature:

Specify your nomenclature of the „sulfur clusters“. You are very unclear, if this really refers to the uncharged neutral clusters (for which the +30 allotropes are known), or if this holds for the ionic clusters that may also directly be characterized by MS. Here again, one has to differentiate between the sulfur anions (which are copious and dianions as well as radical anions have been well characterized in the condensed phases as well as with MS) and sulfur cations. The latter are very scarce in condensed phases and only 3 (S4 (2+), S8 (2+) and S19(2+)) have been crystallized and its structures are known, while in solution as well S5 + , S7 + and S6 (2+) were tentatively assigned (cf.: Inorg. Chem. 2004, 43, 1000-1011). By contrast, the chemistry of sulfur cations in the gas phase as investigated by MS and supported by QM calculations is well known.

Thus, apparently your major claim refers towards the gaseous neutrals, which itself (without host guest chemistry) cannot be investigated by MS. In the condensed phase, the neutrals itself are well studied. Most informative are Raman spectra, but also HPLC and GC have been used to determine the ratio of the constituents of mixtures.

Overall, the picture presented in the manuscript is too black-and-white. I would ask to rephrase.

Abstract: You state: „otherwise labile S6 and S12 clusters“ Hm, this is very relative. They are both actually rather stable and can be prepared as solids (as you have done for S6). S12 even has a higher melting point as S8. Overall S6 and S12 are the two second most stable allotropes following S8, you can prepare them in larger scale and store them as a solid for very extended periods of time at r.t.

Thus, I would ask to rephrase.

Incomplete characterization or missing discussion: S8 is the standard material used to calibrate each commercial Raman spectrometer. It has extremely intense and informative Raman modes; the same holds for S6 and S12. Thus, I would request that you record Raman spectra of your host-guest structures and briefly discuss the shift of the sulfur cluster Raman bands in comparison to those of the free molecules (they are known for all three: S8, S6 and S12). One may substantiate your bonding claims by this on the very sensitive vibrational frequency shift scale. Therefore, spectra of the S6 and S12 complexes included with Figure 4c need some more evaluation and discussion. There are more bands visible than those given, be more specific. One can learn a lot from those bands. Place the bands of the isolated neutrals above the recorded spectra of your encapsulated materials to see the relation and which bands belongs to the sulfur clusters.

Figure 2: The caption and the numbering scheme in the figure do not match. Thus, the MALDITOF-Spectrum of S8 is labelled f) in the figure and e) in the caption. Again the MALDI-TOF-MS does not show neutral Sulfur clusters, but rather sulfur ions. Please specify which. I would assume the cation mode, as the sulfur cations shown are clearly the typical reaction products of ionization of neutral S8 giving the typical range of most stable sulfur monocations as a consequence of the ionization procedure.

I realize that the authors are not aware that this degradation process of the initially formed S8 + radical cation is in effect thermodynamically driven by the dismutation to the most stable S5 + and S7 + cations (and also notable amounts of S3 +). The enthalpies of formation of the gaseous cluster cations have been published and the reaction that is depicted in Figure 2 (e or f), (S8 + = S5 + + neutral, e.g. S2 or S3 and related) is the intrinsic chemistry of sulfur cations (but not neutrals).

Thus: You have developed a unique method to study the neutrals and observe their reaction. This can only be done like this, but the comparison to the study of the free clusters is not valid as such, as the species investigated are different (charged vs. neutral). So you are comparing apples with pears.

The reaction of encapsulated 2 S6 giving S12: This is a nice reaction, but again in total agreement with thermodynamics. Out of all allotropes of sulfur, S12 is the second most stable (after S8). Thus, the transformation of 2 S6 = S12 is a downhill process initiated by Vis irradiation.

Note: that with your LED Lamp at 425+/-15 nm you remain visible, not UV. Please note the thermodynamics in your contribution and add a short sentence or two.

Typos: P3: Add „solution“ after „D2O (0.5 mL)“

For the comments of Reviewer #1:

The manuscript entitled “A Polyaromatic Capsular Matrix: Exact Mass Analysis of Sulfur Clusters upon Encapsulation” by Yoshizawa coworkers reported the stability of some Sn clusters (S6, S8) under ESI-MS condition when they are encapsulated by a Pd-cage. This manuscript cannot be recommended to Nature Commun for publication due to the following literature precedence and reviewer’s analysis.

We would like to emphasize the novelty and significance of this report by the following response to the comments of Reviewer #1.

1) Synthesis of the cage was previously demonstrated by the authors (N. Kishi, Z. Li, K. Yoza, M. Akita and M. Yoshizawa, J. Am. Chem. Soc., 2011, 133, 11438-11441). Following, encapsulation of certain disulfide molecules by the same cage is also reported by them (N. Kishi, Z. Li, Y. Sei, M. Akita, K. Yoza, J. S. Siegel and M. Yoshizawa, Chem. Eur. J., 2013, 19, 6313-6320).

Whereas we have already reported the synthesis and host properties of polyaromatic capsule **1**, the reported guests are limited to only *organic* molecules (including *organic* dithia[3.3]paracyclophane) (ref. 18, 23, 24). In marked contrast, we as well as other groups have not succeeded in the encapsulation of *inorganic* sulfur clusters by molecular capsules or cages *in solution* before this report.

2) Stabilization of reactive species by the same cage. Authors have already shown radical initiator AIBN ion could be stabilized towards light and heat by the same Pd-Cage (M. Yamashina, Y. Sei, M. Akita and M. Yoshizawa, Nature Commun., 2014, 5, 4662).

There are many reports on the stabilization of reactive *organic* species by molecular cages and capsules (ref. 8-18). On the other hand, except for only Nitschke’s work in 2009 (ref. 19), there has been no report on the stabilization of reactive *inorganic* clusters within molecular cages. Stabilization of inorganic clusters upon encapsulation is not common but still very rare.

3) Literature on encapsulation of S8. Raptis and co-workers have reported the crystal structure and stability of S8 with trigonal prismatic cage molecules (P.-C. Duan, Z.-Y. Wang, J.-H. Chen, G. Yang and R. G. Raptis, Dalton Trans., 2013, 42, 14951-14954).

The Raptis’s paper (ref. 26) reported the crystal structure and *solid-state* thermal properties of an S₈-cage complex. There are no structural data of the product *in solution*. In contrast, our studies focus on the subject of “*mass determination of the uncharged sulfur clusters have virtually failed owing to the instability under usual electron ionization conditions. Therefore, no reliable method to obtain the structural information of such metastable clusters in solution has been reported.*” Only our method can obtain reliable MS data from sulfur clusters.

4) Literature on stabilization of S6. Sugimoto et al., reported a discovery of air-stable S6 molecules in the solid state via crystallization of co-crystals with 3,5-diphenyl-1,2,4-dithiazol-1-ium (dpdti) iodide (ref: K. Sugimoto, H. Uemachi, M. Maekawa and A. Fujiwara, Cryst. Growth Des., 2013, 13, 433-436). The ring-structure cyclo-S6 molecule has been reported to be stored in cocrystals for at least half a year. Contrastingly authors have reported here identification of <10% of the cage⊃(S6•S8) complex at NMR time scale.

The Sugimoto’s paper reports on the stabilization of S₆ in the *co-crystal state* (ref. 33). However, there has been no report on the stabilization *in solution* so far. Our capsule can significantly stabilize S₆ *in solution* as well as facilitate the MS identification of S₆. More importantly, we succeeded in *in situ* preparation of a reactive S₁₂ cluster from two S₆

clusters within the capsule. To the best of our knowledge, there is no report on these unusual behaviors.

This findings can thus be rationalized as a routine extension to the concept of cage to stabilization of reactive species (references 8-19). As a reviewer, I am not convinced that the results signify a sufficiently notable advance to justify publication in Nature Communications.

As mentioned above, the present “encapsulation”, “stabilization”, and “MS characterization” and “*in situ* preparation” of inorganic sulfur clusters are undoubtedly novel. Besides the evaluations by other reviewers, we believe that the present results are novelty and significance enough for publication in *Nature Commun.*

For the comments of Reviewer #2:

This well-written and well-illustrated manuscript describes the preparation and characterization of three supramolecular compounds (in tiny quantities) which should be highly interesting to both main-group and coordination chemists. In principle, water-insoluble elemental sulfur has been encapsulated in aqueous solution by a polyaromatic palladium complex 1 characterized by a large inner cavity which is big enough to host two S₈ or two S₆ molecules resulting in the crystalline species 1x(S₈)₂ and 1x(S₆)₂. These two species have been well characterized spectroscopically as well as by X-ray structural analysis. It is obvious that these findings will be the starting point for many subsequent studies on the encapsulation and stabilization of other hydrophobic inorganic molecules. A third compound claimed to be 1x(S₁₂) has been obtained by low-temperature irradiation of 1x(S₆)₂. However, this compound has been less well characterized (no X-ray structure analysis) and should be either omitted from the manuscript or better characterized. Especially, since the Raman spectrum of 1x(S₁₂) does not show the most intense line of S₁₂ at 128 cm⁻¹. A better spectrum would be of great help to convince this reviewer.

Thus, I strongly recommend acceptance of the manuscript for publication after additional revisions as follows:

We appreciate having very positive evaluation from the Reviewer 2 on this work.

Regarding the characterization of the S₁₂ cluster within the capsule, we have obtained the following four experimental data; (i) the ¹H NMR spectrum indicates the selective formation of a new cluster (except for S₆ and S₈) possessing relatively high symmetry, (ii) the photoproduct could not be extracted from the capsule because of the bulkiness, whereas encapsulated small clusters (e.g., S₆ and S₈) are easily extracted by a typical extraction protocol, (iii) the ESI-TOF MS spectrum establishes the encapsulation of a 12•S component within the capsule, and (iv) the Raman spectrum shows new peaks assignable to a S₁₂ cluster at 445 and 457 cm⁻¹. We collaborated with Dr. Akiyoshi Kuzume (coauthor), who is majoring in Raman spectroscopy, to carefully evaluate the Raman peaks for the otherwise labile S₁₂ cluster. A peak at ~130 cm⁻¹ could not be observed in the Raman spectrum (Supplemental Fig. 22) because of intense broad peaks (<180 cm⁻¹) derived from the capsule. However, combined with the obtained ¹H NMR and ESI-TOF MS data, and the extraction study, the observation of two Raman peaks at 446 and 458 cm⁻¹ (the intensity of the latter peak is usually comparable to that at ~130 cm⁻¹) undoubtedly supports the formation of the S₁₂ cluster.

In addition, theoretical study on the S₁₂-capsule complex by force-field calculations also supports the host-guest structure of the product (Fig. 4e). We added the proposed

mechanism of the S_{12} formation to the revised text as follows: "... the cyclic S_6 cluster provides a highly strained ring structure. Thus, the formation of the S_{12} cluster through the homo-coupling of two biradical S_6 species (Fig. 4a, right), generated by homolytic photocleavage of the S-S bonds of $(S_6)_2$, might be a thermodynamically favorable process."

1) correction of ref. 31 (two spelling errors).

The typos were corrected.

2) Also, 12 times the word "exact" is used in the manuscript which seems somewhat exaggerated since measurements should always be exact.

According to the reviewer's suggestion, we removed most of the "exact" from our manuscript.

3) Furthermore, MS is not a method for structural characterization of sulfur clusters (page 1) but just for mass determination; the most important spectroscopic technique for sulfur clusters is Raman spectroscopy which can also be applied to solutions (page 2).

We replaced "MS characterization" by "mass determination" in the introduction. Most of sulfur clusters (except for S_8) are unstable *in solution* and their solubilities are usually poor so that it is quite difficult to obtain clear Raman peaks from the solutions. Actually, it is reported that "The solution (Raman) spectra were extremely weak because of the rapid conversion of S_6 to S_8 (J. Molecular Spectroscopy 22, 105 (1967))" and "Because of the very slight solubility of S_{12} in suitable solvents and the decomposition of S_{12} at the melting point, only solid state spectra could be measured (J. Molecular Spectroscopy 51, 189 (1974))".

4) The closest distances between the sulfur atoms and the eight anthracene panels of 3.6 Å (page 4) hardly suggest the presence of any S- π interactions and the hydrophobic effect is most probably responsible alone for the formation of these encapsulated sulfur complexes.

The 3.6 Å is slightly longer than the sum of the van der Waals radius of sulfur and carbon atoms (3.50 Å). According to the reviewer's suggestion, we removed "which suggest the presence of S- π interactions" from the corresponding sentence.

5) On p. 6 the authors should mention whether or not the slow decomposition of $1x(S_6)_2$ in aqueous solution results in the increase of signals due to $1.(S_6.S_8)$ or what other decomposition products have been observed; see Fig. 3e.

We added the following sentence to the corresponding paragraph: "... and the ESI-TOF MS analysis of the resultant solution indicated the generation of $1\mathcal{D}(S_6\bullet S_8)$ and $1\mathcal{D}(S_6\bullet S_7)$ species (Fig. 3c, right)".

For the comments of Reviewer #3:

The ms by Yoshizawa et al reports a very appealing mass spectrometry and single crystal X-ray diffraction (SCXRD) study on sulfur S_6 , S_8 and S_{12} clusters captured inside a nanocontainer. As the nanocontainer they have designed an ingeniously simple polyaromatic nanocapsule which is then used a supramolecular matrix. The sulfur clusters fragment under usual mass spectrometry conditions, yet isolation and protecting the S-cluster inside the nanocontainer provides stable 1:2 host-guest complexes for a routine ESI-TOF MS analysis. The cationic and fully closed polyaromatic nanocontainer not only protects the clusters, but also facilitates the efficient ionization of the clusters without fragmentation. In addition, the capsular matrix show a promise that they could be used as nanoreactors. The authors have also provided SCXRD proof about the encapsulated S-clusters, this is not an easy task yet they have done it remarkably well. I am very happy to

support the publication of this excellent piece of work in Nature Communications, provided that some minor matters in the checkcif files are commented, see below.

We greatly appreciate having very positive evaluation from the Reviewer 3 on all of this work (except for minor alerts on the checkCIF report).

1) The checkcif report contain quite a bit A and B level alerts. This is natural with such complex X-ray structures, yet the authors should comment these alerts using the IUCr Output Validation Response Form available from <http://checkcif.iucr.org/>

We would like to thank this reviewer for this helpful comment. We have already prepared “Refinement details of X-ray crystallographic analysis” sections in the SI (page 14 and 15) and additionally added responses to these alerts.

For the comments of Reviewer #4:

This is a very interesting paper that reports on an interesting method that might be uniquely suited to study neutral main group clusters in the gas phase that would otherwise be charged and undergo thermodynamically induced mainly dismutation reactions. Thus, I can support publication, yet only after the below addressed major concerns are addressed.

We appreciate having positive evaluation and useful comments from the Reviewer 4 on this work.

Remarks: Overall the authors use the study of neutral and charged gaseous clusters with absolutely no differentiation. This is not valid. The uniqueness of their method is that it allows to study neutral clusters encapsulated into a matrix of an MOF-like environment. In addition, they give absolutely no thermodynamic reasoning for the reactivity observed (i.e. the dimerization of 2 S₆ giving S₁₂). However, this is a downhill reaction and only logic. Thus, the driving force needs to be accounted for.

We fully agree with two reviewer’s comments described above. We need to add “neutral” to clarify this work. The S-S bonds of sulfur clusters are photochemically (as well as thermally) labile and thus reversibly generate radical species upon light irradiation through the homolysis of the S-S bonds (e.g., See Fig. 4a, right). Cyclic S₆ cluster provides a highly strained ring structure. Therefore, only in the confined cavity of the capsule, two S₆ clusters can selectively transform into one S₁₂ cluster, which is a less strained, crown-shaped molecule, upon light irradiation. This is a thermodynamically favorable, downhill reaction, as mentioned by this reviewer.

I have addressed all those points following their appearance in the text in the following. Once those points are addressed, and suitably incorporated in a revised manuscript with re-review, the manuscript may become suitable for NCOMMS.

We hope that the following our responses and revisions are suitable for publication in *Nature Communications*.

Nomenclature:

1) Specify your nomenclature of the „sulfur clusters“. You are very unclear, if this really refers to the uncharged neutral clusters (for which the +30 allotropes are known), or if this holds for the ionic clusters that may also directly be characterized by MS.

According to these reviewer’s comments, we added “neutral” and “uncharged” to the revised text.

2) Here again, one has to differentiate between the sulfur anions (which are copious and dianions as well as radical anions have been well characterized in the condensed phases as

well as with MS) and sulfur cations. The latter are very scarce in condensed phases and only 3 ($S_4(2+)$, $S_8(2+)$ and $S_{19}(2+)$) have been crystallized and its structures are known, while in solution as well S_5+ , S_7+ and $S_6(2+)$ were tentatively assigned (cf.: *Inorg. Chem.* 2004, 43, 1000-1011). By contrast, the chemistry of sulfur cations in the gas phase as investigated by MS and supported by QM calculations is well known.

We focus on neutral sulfur clusters in this report.

3) Thus, apparently your major claim refers towards the gaseous neutrals, which itself (without host guest chemistry) cannot be investigated by MS. In the condensed phase, the neutrals itself are well studied. Most informative are Raman spectra, but also HPLC and GC have been used to determine the ratio of the constituents of mixtures.

We would like to emphasize the stabilization effect of our capsular matrix. In addition, tiny quantities (microgram scale) of samples, even including several known and/or unknown compounds, are enough to prepare the host-guest composites for mass determination.

4) Overall, the picture presented in the manuscript is too black-and-white. I would ask to rephrase.

We had already used many colors in all of the figures but we used more colors in the revised manuscript.

5) Abstract: You state: „otherwise labile S_6 and S_{12} clusters“ Hm, this is very relative. They are both actually rather stable and can be prepared as solids (as you have done for S_6). S_{12} even has a higher melting point as S_8 . Overall S_6 and S_{12} are the two second most stable allotropes following S_8 , you can prepare them in larger scale and store them as a solid for very extended periods of time at r.t. Thus, I would ask to rephrase.

We have described that, under not only MS conditions but also ambient aqueous conditions, S_6 and S_{12} clusters are labile. The stabilities of S_6 and S_{12} in solution state are very poor. To avoid this misleading, we had mentioned “Free S_6 dissolved in CS_2 was completely decomposed into S_8 and other oligosulfurs within ~1 h at room temperature (Fig. 3e)” in the text.

6) Incomplete characterization or missing discussion: S_8 is the standard material used to calibrate each commercial Raman spectrometer. It has extremely intense and informative Raman modes; the same holds for S_6 and S_{12} . Thus, I would request that you record Raman spectra of your host-guest structures and briefly discuss the shift of the sulfur cluster Raman bands in comparison to those of the free molecules (they are known for all three: S_8 , S_6 and S_{12}).

According to these reviewer’s comments, we added the Raman data of S_{12} within/without the capsule to the main text as follows: “The observed new peaks at 445 and 457 cm^{-1} are overlapped with the characteristic peaks of free S_{12} (i.e., 445 and 456 cm^{-1} , respectively)”. Furthermore, “On the other hand, Raman peaks derived from clusters S_6 (e.g., 200 and 264 cm^{-1} ; Fig. 4c, middle) and S_8 (e.g., 149 and 218 cm^{-1}) within **1** were virtually undetected in the spectrum. These peaks are slightly shifted (up to $-4 cm^{-1}$) as compared with those of free S_6 and S_8 (See Supplementary Figs 16 and 9).” were also added to the main text.

7) One may substantiate your bonding claims by this on the very sensitive vibrational frequency shift scale. Therefore, spectra of the S_6 and S_{12} complexes included with Figure 4c need some more evaluation and discussion. There are more bands visible than those given, be more specific. One can learn a lot from those bands. Place the bands of the isolated neutrals above the recorded spectra of your encapsulated materials to see the relation and which bands belongs to the sulfur clusters.

There are many strong and broad Raman peaks derived from the empty capsule (Fig. 4c, top). Except for the typical Raman peaks for S_6 (200 and 264 cm^{-1}) and S_{12} (445 and 457 cm^{-1}), it is quite difficult to discuss other peaks found in the Raman spectra. Thus, we added the values of the characteristic peaks to the Raman spectra in the revised SI. In contrast, the ^1H NMR signal of inner protons H_f is very sensitive to the guest molecule inside the capsule. We added the following sentence to evaluate the effective capsule-cluster interactions: “*In addition, a huge upfield shift of inner protons H_f is indicative of the encapsulation of a bulky cluster*”.

8) *Figure 2: The caption and the numbering scheme in the figure do not match. Thus, the MALDITOF-Spectrum of S8 is labelled f) in the figure and e) in the caption. Again the MALDI-TOF-MS does not show neutral Sulfur clusters, but rather sulfur ions. Please specify which. I would assume the cation mode, as the sulfur cations shown are clearly the typical reaction products of ionization of neutral S8 giving the typical range of most stable sulfur monocations as a consequence of the ionization procedure.*

I realize that the authors are not aware that this degradation process of the initially formed $S8+$ radical cation is in effect thermodynamically driven by the dismutation to the most stable $S5+$ and $S7+$ cations (and also notable amounts of $S3+$). The enthalpies of formation of the gaseous cluster cations have been published and the reaction that is depicted in Figure 2 (e or f), ($S8+ = S5+ + \text{neutral}$, e.g. $S2$ or $S3$ and related) is the intrinsic chemistry of sulfur cations (but not neutrals).

The wrong caption and the figure of the MALDI-TOF MS spectrum were corrected.

9) *Thus: You have developed a unique method to study the neutrals and observe their reaction. This can only be done like this, but the comparison to the study of the free clusters is not valid as such, as the species investigated are different (charged vs. neutral). So you are comparing apples with pears.*

We added “*neutral*” and “*uncharged*” to the revised text to clarify the results of this report.

10) *The reaction of encapsulated 2 $S6$ giving $S12$: This is a nice reaction, but again in total agreement with thermodynamics. Out of all allotropes of sulfur, $S12$ is the second most stable (after $S8$). Thus, the transformation of $2 S6 = S12$ is a downhill process initiated by Vis irradiation.*

We added the following sentences to the revised manuscript to explain this phenomenon: “*Concerning the unusual clusterization in the isolated nanospace, the cyclic S_6 cluster provides a highly strained ring structure. Thus, the formation of the S_{12} cluster through the homo-coupling of two biradical S_6 species (Fig. 4a, right), generated by homolytic photocleavage of the S-S bonds of $(S_6)_2$, might be a thermodynamically favorable process.*”

11) *Note: that with your LED Lamp at 425 \pm 15 nm you remain visible, not UV. Please note the thermodynamics in your contribution and add a short sentence or two.*

We added “*This photoreaction occurred very slowly under UV light irradiation ($\lambda_{\text{irrd}} = 360$ nm), probably due to the strong absorption bands (~ 380 nm) of the polyaromatic capsule*” to the revised manuscript.

12) *Typos: P3: Add „solution“ after „D2O (0.5 mL)“*

The typo was corrected.

Reviewers' comments:

Reviewer #2 provided comments to the editor only, in which he/she recommended publication.

Reviewer #3 (Remarks to the Author):

All my concerns about the checkcif alerts have been fully explained in the SI and I am happy to support the ms in its present form.

Reviewer #4 (Remarks to the Author):

The current revision improved, but uses in part a very imprecise and sometimes clearly wrong scientific language. This definitely has to be addressed in a revised version with re-review prior to publication.

I guess the main point is that the authors are by training no inorganic main group chemists but touch with their beautiful method this area. Any inorganic chemist reading this article would not accept several major statements that need to be tackled.

Here are points of concern that need to be changed:

Abstract: The statement that characterization of neutral sulfur clusters by MS "...has practically failed due to their instability under electron ionization conditions." is wrong.

I have the impression this is not clear to you:

Oligosulfur anions and cations (as intrinsically produced by any MS method) are very different to neutrals. And charged sulfur clusters have been extensively studied in the MS. And most of those have been prepared by ionization of the neutrals or by LDI of polysulfides.

Thus not the MS analysis of neutral oligosulfurs has failed, but naturally the obtained particles are different. You produce ions with a distinctly different chemistry.

Thus: this sentence must be removed, it is simply wrong.

Again: What you have developed is the method to directly study the neutrals by MS. This is the key point, forget about the rest.

In addition: Is the "the mass determination" really the key point...? In your capsule, you cannot only study the mass, but also reactions. And you have a means in hand to correlate those results also to the condensed phase. This is unique.

See your Fig. 1f. Although the signal for S₈⁺ is tiny, it would be sufficient for mass determination, wouldn't it...?

Please change the text.

Intro: Your statement "However, most of the neutral clusters are unstable under MS conditions", now using neutral, is not correct as such.

The neutral clusters (S₈, P₄, etc.) may without problem be sublimed in preparative quantities.

However, in the MS upon ionization they are unstable with respect to other fragmentation products

and therefore fragment. I.e. even S_8^+ preferentially transforms to uneven S_n^+ ($n = 3, 5, 7$, also 4 (the reason is, that S_4 neutral is very unstable and therefore easy to oxidize)). The same holds for P_4^+ (a radical), which is less stable than P_3^+ (closed shell) and fragments.

Please change the text.

In alignment with this: The text "yet mass determination of the uncharged structures has virtually failed owing to the instability under usual electron ionization conditions" needs to be exchanged.

What you mean is that "yet mass determination of the uncharged metastable structures was hitherto virtually impossible by MS methods. Owing to the instability of the under usual electron ionization conditions formed oligosulfur ions, immediate and substantial fragmentation reactions were typically observed (Fig. 1a, left)."

Please change the text.

Above Fig. 1: Your statement: Nevertheless, those of labile inorganic clusters are limited to white phosphorus (Fig. 1d) reported by the Nitschke group.¹⁹

Hm, I would check the literature for the work of Manfred Scheer et al in Regensburg. Using a supramolecular approach (but less common, inorganic ligands) they have encapsulated a lot more, including C_{60} , As_4 and more. Published up to Science. Please give them a reference.

p. 5: Replace "labile sulfur allotrope" by "metastable sulfur allotrope"

p. 5: Please rephrase: "It is worth noting that the otherwise labile S_6 cluster, which possesses highly strained S-S bonds,"

Hm, I was always told that the strain in a six membered ring in a chair conformation is minimized, isn't it...?

So it is not the strain, but the intrinsic lower S-S bond energy (thermodynamics) in combination with lower activation energies (kinetics) that transform the metastable S_6 to the stable S_8 molecule.

Please rephrase.

p. 7, line 4: In the caption to Figure 3c, you address this as only $1@S_6^*S_8$, but here you state also $1@S_6^*S_7$. I suppose, this belongs to the smaller signal envelope in the middle. This makes it even more interesting, as S_7 is pretty much the least stable sulfur allotrope and from the bonding the by far most interesting.

Why is S_7 formed here. This is a pretty uphill reaction... Fascinating. Can you comment on...?

p.8, first yellow part. Look at my above comment with respect to strain. Change similarly as the S_6 S_8 story. Refer to the known enthalpies of formation in gas and solid phase. S_{12} is more stable than 2 S_6 . Refer to stability (thermodynamics) and include activation energy (kinetics) overcome by irradiation.

Fig. 4a, right. It is highly unlikely that you will have two diradicals sitting next to each other. The activation energy for this process is already overcome by formation of one S_6 diradical, which then will immediately react with the second intact S_6 molecule close by. This is classical sulfur chemistry and takes place (thermally induced) in sulfur melts at temperatures above 150 °C (vast literature). Please redraw the Picture. Sulfur mechanisms are different...!

The rest is fine.

For the comments of Reviewer #2:

Reviewer #2 provided comments to the editor only, in which he/she recommended publication.

We appreciate having the recommendation of the publication from the Reviewer 2.

For the comments of Reviewer #3:

All my concerns about the checkcif alerts have been fully explained in the SI and I am happy to support the ms in its present form.

We also appreciate having the support from the Reviewer 3 for the publication.

For the comments of Reviewer #4:

The current revision improved, but uses in part a very imprecise and sometimes clearly wrong scientific language. This definitely has to be addressed in a revised version with re-review prior to publication.

We would like to thank the Reviewer 4 for his/her kind comments to improve our manuscript.

I guess the main point is that the authors are by training no inorganic main group chemists but touch with their beautiful method this area. Any inorganic chemist reading this article would not accept several major statements that need to be tackled.

Here are points of concern that need to be changed:

1) Abstract: The statement that characterization of neutral sulfur clusters by MS "...has practically failed due to their instability under electron ionization conditions." is wrong.

I have the impression this is not clear to you:

Oligosulfur anions and cations (as intrinsically produced by any MS method) are very different to neutrals. And charged sulfur clusters have been extensively studied in the MS. And most of those have been prepared by ionization of the neutrals or by LDI of polysulfides.

Thus not the MS analysis of neutral oligosulfurs has failed, but naturally the obtained particles are different. You produce ions with a distinctly different chemistry.

Thus: this sentence must be removed, it is simply wrong.

According to the reviewer's comment, we removed "... has practically failed due to their instability under electron ionization conditions" and mentioned the general finding "MS analysis of oligosulfurs (S_n) ... usually displays their fragment peaks" in the revised abstract.

2) In addition: Is the "the mass determination" really the key point...? In your capsule, you cannot only study the mass, but also reactions. And you have a means in hand to correlate those results also to the condensed phase. This is unique.

In the Discussion session, we have also emphasized "The present functions as an analytical tool as well as a reaction vessel for sulfur clusters...". We developed a new MS method and thereby we could determine the product of the photoreaction within the capsule. Therefore, we concluded that "mass determination" is the most important result in this report.

- 3) See your Fig. 1f. Although the signal for S₈⁺ is tiny, it would be sufficient for mass determination, wouldn't it...?

Please change the text.

In Fig. 2h, we could find a tiny peak for a S₈⁺ species with many prominent fragment peaks derived from S₈. However, MS spectrum of S₆ (Supplementary Fig. 15) is quite similar to that of S₈ under the same analytical conditions. Therefore, it is very difficult to determine the product composition (and also the purity) of sulfur clusters S_n by common MS analysis.

- 4) Intro: Your statement "However, most of the neutral clusters are unstable under MS conditions", now using neutral, is not correct as such.

The neutral clusters (S₈, P₄, etc.) may without problem be sublimed in preparative quantities. However, in the MS upon ionization they are unstable with respect to other fragmentation products and therefore fragment. I.e. even S₈⁺ preferentially transforms to uneven S_n⁺ (n = 3, 5, 7, also 4 (the reason is, that S₄ neutral is very unstable and therefore easy to oxidize)). The same holds for P₄⁺ (a radical), which is less stable than P₃⁺ (closed shell) and fragments.

Please change the text.

According to the reviewer's comment, we replaced "However, most of the neutral clusters are unstable under MS conditions..." by "However, most of the neutral clusters fully or partially decompose within MS ..." in the revised text.

- 5) In alignment with this: The text "yet mass determination of the uncharged structures has virtually failed owing to the instability under usual electron ionization conditions" needs to be exchanged.

What you mean is that "yet mass determination of the uncharged metastable structures was hitherto virtually impossible by MS methods. Owing to the instability of the under usual electron ionization conditions formed oligosulfur ions, immediate and substantial fragmentation reactions were typically observed (Fig. 1a, left)."

Please change the text.

According to the reviewer's helpful suggestion, We exchanged "... yet mass determination of the uncharged structures has virtually failed owing to the instability under usual electron ionization conditions" for "..., yet mass determination of the uncharged structures has been virtually impossible by previous MS methods. Owing to the instability of oligosulfur ions generated under usual electron ionization conditions, immediate and substantial fragmentation reactions are typically observed (Fig. 1a, left):".

- 6) Above Fig. 1: Your statement: Nevertheless, those of labile inorganic clusters are limited to white phosphorus (Fig. 1d) reported by the Nitschke group.¹⁹

Hm, I would check the literature for the work of Manfred Scheer et al in Regensburg. Using a supramolecular approach (but less common, inorganic ligands) they have encapsulated a lot more, including C₆₀, As₄ and more. Published up to Science. Please give them a reference.

We overlooked the reports on the stabilization of an As₄ cluster (yellow arsenic). We added the related Scheer's and Wu's papers to the Reference section (ref. 20 and 21) and

modified the related sentence (page 2).

7) p. 5: Replace "labile sulfur allotrope" by "metastable sulfur allotrope"

The "labile" was replaced by "metastable".

8) p. 5: Please rephrase: "It is worth noting that the otherwise labile S6 cluster, which possesses highly strained S-S bonds,"

Hm, I was always told that the strain in a six membered ring in a chair conformation is minimized, isn't it...?

So it is not the strain, but the intrinsic lower S-S bond energy (thermodynamics) in combination with lower activation energies (kinetics) that transform the metastable S6 to the stable S8 molecule.

Please rephrase.

Structural features of cyclic sulfur clusters are different from those of cyclic alkanes. The S-S distances of S₆ (2.057 Å) are comparable to those of S₈ (2.060 Å) (and also S₁₂). On the other hand, the torsion angles of S₆ (ca. 74°) are quite different from those of S₈ (ca. 98°) (ref. 4, page 374). The narrow torsion angles increase the repulsion between lone pair electrons on adjacent sulfur atoms. Therefore, "which possesses highly strained S-S bonds" was rephrased by "which possesses a highly strained ring structure", which is also supported by the following sentences from ref. 6 (page 50): "... Since the torsion angles in S₆ (D_{3d} symmetry) are all equal and ca. 74° which is an unfavorable value, the strain is "stored" in the whole ring."

9) p. 7, line 4: In the caption to Figure 3c, you address this as only 1@S6*S8, but here you state also 1@S6*S7. I suppose, this belongs to the smaller signal envelope in the middle. This makes it even more interesting, as S7 is pretty much the least stable sulfur allotrope and from the bonding the by far most interesting.

Why is S7 formed here. This is a pretty uphill reaction... Fascinating. Can you comment on...?

Now we have not other evidence of the formation of S₇ except for the ESI-TOF MS data. ESI-TOF MS spectrum of 1⊃(S₈)₂ does not show such MS peaks (Fig. 2i). Capsule 1 can selectively bind two S₈ clusters from a mixture of S₈ and S₆ clusters due to size-complementary host-guest interactions. References 33 and 36 report that S₇ is more stable than S₆ from the viewpoints of reaction enthalpy (S₇ < S₆ (ΔH = +1 kJ/mol)) and theoretical binding energy (S₇ > S₆). On the basis of these findings, one proposed reason is that the stabilities of S₆ and S₇ in the confined cavity of 1 are different from those in bulk solutions: S₇ is more stable than slightly smaller S₆ in the cavity. Actually, we did not observe MS peaks derived from much smaller S₅ and S₄ in Fig. 3c.

10) p.8, first yellow part. Look at my above comment with respect to strain. Change similarly as the S6 S8 story. Refer to the known enthalpies of formation in gas and solid phase. S12 is more stable than 2 S6. Refer to stability (thermodynamics) and include activation energy (kinetics) overcome by irradiation.

S₁₂ is the next most stable sulfur allotrope after S₈ but gradually decomposes into S₈ and S_n oligomers *in solution* even at room temperature. The average torsion angle of S₁₂ (86°) is wider than that of S₆ (ca. 74°) but narrower than that of the most stable S₈ (ca. 98°). It is quite difficult to discuss all of the reasons why S₆ is so unstable but the ring strain is one of the major reasons. According to the reviewer's comment, we added two papers to

ref. 33 and 36, which report the reaction enthalpies ($S_8 < S_{12}$ ($\Delta H = +12$ kJ/mol) $< S_6$ ($\Delta H = +22$ kJ/mol)) and the theoretical binding energies per atom of the most stable S_n isomer ($S_8 \approx S_{12} > S_6$), respectively. We would like to discuss the details after collaboration studies with theoretical experts using our host-guest structures in another paper.

11) Fig. 4a, right. It is highly unlikely that you wil have two diradicals sitting next to each other. The activation energy for this process is already overcome by formation of one S6 diradical, which then will immediately react with the second intact S6 molecule close by. This is classical sulfur chemistry and takes place (thermally induced) in sulfur melts at temperatures above 150 °C (vast literature). Please redraw the Picture. Sulfur mechanisms are different...!

According to the helpful reviewer's comment, we redrew the Fig. 4b and modified the related sentence.

: REVIEWERS' COMMENTS:

Reviewer #4 (Remarks to the Author):

The revised manuscript greatly improved and I do have only a few comments, that can be addressed in a minor revision without re-review.

The only scientific point relates to this sentence:

"It is worth noting that the otherwise labile S6 cluster, which possesses highly strained ring structure"

a) If S6 would be labile, one could not crystallize it and obtain single crystals suitable for XRD. Thus it is metastable with a low barrier for fragmentation. This is what I found as a definition for lability: "The term is used to describe a transient chemical species. As a general example, if a molecule exists in a particular conformation for a short lifetime, before adopting a lower energy conformation (structural arrangement), the former molecular structure is said to have 'high lability'"

I guess, this is not what you want to say. It is metastable with low barrier.

b) I insist with the strain. A six membered ring in chair conformation as such has no "strain". However, it is the unfavorable S-S-S-S torsion angle that leads to repulsion of the 3p²-type lone pair orbitals (ideal 90°, worst 0° as in one case for S7. This leads to an enlargement of the central S-S bond in S7 to 218 pm (cf. 205 pm in S8). In addition you do have the 1,3-lone pair repulsion that leads to an increase of the S-S-S bond angle. With s-p-separation the typical and electronically preferred S-S-S bond angle would be 90° (as approximately in H₂S). However, the 1,3-lone pair orbitals come close and lone pair repulsion leads to an increase of the electronically by the participating orbitals preferred 90° arrangement.

Thus, essentially it is 1,3- and 1,4-lone pair repulsion.

Therefore I do like the inclusion of bond angles in the discussion, but again: Strain is not the right expression, when dealing with a six membered ring in chair conformation, as it will be misunderstood.

I would be happy if you replace "strain" by "sulfur lone pair repulsion".

For the comments of Reviewer #4:

The revised manuscript greatly improved and I do have only a few comments, that can be addressed in a minor revision without re-review.

We appreciate having very positive evaluation from Reviewer #4 on our revised manuscript.

"It is worth noting that the otherwise labile S6 cluster, which possesses highly strained ring structure"

a) If S6 would be labile, one could not crystallize it and obtain single crystals suitable for XRD. Thus it is metastable with a low barrier for fragmentation. This is what I found as a definition for lability: "The term is used to describe a transient chemical species. As a general example, if a molecule exists in a particular conformation for a short lifetime, before adopting a lower energy conformation (structural arrangement), the former molecular structure is said to have 'high lability'"

I guess, this is not what you want to say. It is metastable with low barrier.

Ref. 32 reported that the crystals of S6 were prepared from the CS2 solutions at $-78\text{ }^{\circ}\text{C}$. In addition, we have already revealed that "Free S6 dissolved in CS2 was completely decomposed into S8 and other oligosulfurs within ~ 1 h at room temperature (Fig. 3g and see Supplementary Fig. 17)" (page 5). Therefore, we would like to use "labile" for free S6.

b) I insist with the strain. A six membered ring in chair conformation as such has no "strain". However, it is the unfavorable S-S-S-S torsion angle that leads to repulsion of the $3p^2$ -type lone pair orbitals (ideal 90° , worst 0° as in one case for S7. This leads to an enlargement of the central S-S bond in S7 to 218 pm (cf. 205 pm in S8). In addition you do have the 1,3-lone pair repulsion that leads to an increase of the S-S-S bond angle. With s-p-separation the typical and electronically preferred S-S-S bond angle would be 90° (as approximately in H2S). However, the 1,3-lone pair orbitals come close and lone pair repulsion leads to an increase of the electronically by the participating orbitals preferred 90° arrangement.

Thus, essentially it is 1,3- and 1,4-lone pair repulsion.

Therefore I do like the inclusion of bond angles in the discussion, but again: Strain is not the right expression, when dealing with a six membered ring in chair conformation, as it will be misunderstood.

I would be happy if you replace "strain" by "sulfur lone pair repulsion".

We carefully re-reviewed the reason for instability of S6 through related several literatures. However, we could find "strain" but not find "sulfur lone pair repulsion" in the literatures. Therefore we would like to use "strain" with several references.